# LEARNING ACROSS THE NOISE SPECTRUM: AN APPROACH TO REINFORCEMENT LEARNING WITH ASYNCHRONOUS SIGNALS

## ABSTRACT

Reinforcement learning (RL) frameworks assume agents receive complete observation vectors at each timestep. However, real-world robotic systems typically operate in environments with asynchronous signals, i.e. sensors that update at different frequencies. We model **asynchronous environments** as an instance of a noise-parameterized family of partially observable Markov decision processes (POMDPs). Our primary contribution, **Learning Across the Noise Spectrum (LANS)**, is a novel strategy that exposes the agent to multiple simulated noise regimes during training, implemented using Soft Actor-Critic (SAC) with recurrent neural networks (RNNs). By sampling different asynchronicity rates, we encourage the development of robust estimators. We prove that LANS acts a *time-aware* regularization term, equivalent to a Jacobian penalty along time-sensitive directions. Experiments on MuJoCo environments with simulated asynchronicity demonstrate that LANS outperforms alternative methods on a variety of tasks—up to a factor of $> 1.5\times$ in some instances—offering a solution for robotic systems that must operate with imperfect sensory information.

## 1 INTRODUCTION

Reinforcement learning (RL) (Sutton & Barto, 1998; Mnih et al., 2015) is an established technology for training agents in environments such as competitive games (Silver et al., 2016; Holcomb et al., 2018; Vinyals et al., 2019), conversational language models (Ouyang et al., 2022; Zhu et al., 2023), and industrial manufacturing (Johannink et al., 2018; Zhang et al., 2022). RL research is grounded on the theory of Markov decision processes (MDPs) (Feinberg & Shwartz, 2012), a formalization in which agents receive complete observation vectors at each timestep—an assumption that does not always hold in real-world deployments. Physical systems, particularly robots, must often operate with sensors that update at different and irregular frequencies, creating what we term **asynchronous environments** (Nebot et al., 1999).

In asynchronous settings, observations are composed of signals from multiple sensors (e.g., cameras, thermometers, accelerometers), each with its own renewal rate. At each timestep, agents receive only a subset of signals, based on which sensors have provided new readings. This poses a challenge for RL methods that assume synchronized observations. Partially observable Markov decision processes (POMDPs) (Krishnamurthy, 2016) provide the framework for addressing incomplete information in RL. While general POMDP techniques are applicable to asynchronous environments, the setting has so far eluded dedicated and systematic treatment.

We formalize asynchronous environments as **asynchronous Markov decision processes (AMDPs)**. Asymptotically, AMDPs behave as POMDPs with a ground-truth state space $S = S_1 \times \cdots \times S_n$ consisting of signals from $n$ sensors. Our key insight is that these environments form a smooth, parameterized family of POMDPs. A noise parameter $\omega$ represents the expected ratio of received signals, ranging from $\omega = 0$ (no signal updates) to $\omega = 1$ (complete readings).

The analysis informs our primary contribution: **Learning Across the Noise Spectrum (LANS)**, an RL learning strategy that exposes the agent to varied noise regimes during training. LANS is a regularization technique that builds on the literature of noise regularization (Bishop, 1995; Sajjadi et al., 2016). Our core novelty is a noising process that acts along the *time axis* and works by

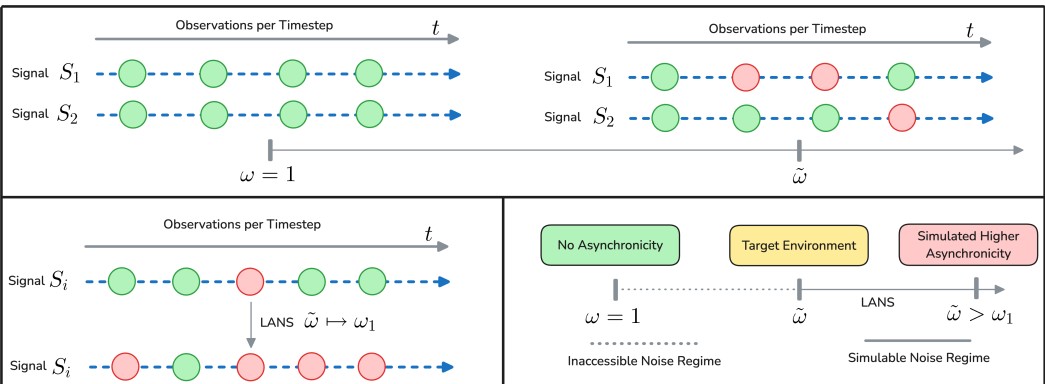

Figure 1: *Top* Parametrization of asynchronous environments by noise $\omega$; lower $\omega$ imply sparser sensor updates. *Bottom-Left* LANS mechanism of noise simulation, randomly masking observed signal during training. *Bottom-Right* High-level illustration of LANS, showing the ability to simulate environments with higher noise $\omega \leq \tilde{\omega}$ than the target one, leading to more robust policies.

simulating environments with higher asynchronicity than that of the target task ($\omega \leq \tilde{\omega}$), as depicted in the *Bottom-Right* of Figure 1. LANS promotes robustness, acting as a Jacobian penalty. We prove that this follows from the loss objective implied by LANS and support it with empirical evidence.

We build a new benchmark suite and framework for defining and evaluating AMDPs. Experiments on asynchronous MuJoCo environments (Tassa et al., 2018) show that LANS significantly outperforms baseline methods, by a factor of $1.5\times$ in some cases. The approach provides a bridge between theory and practice for deploying RL in real-world systems.

Our contributions are threefold:

1. We provide a formalization of **asynchronous Markov decision processes** (**AMDPs**) and prove that they admit an approximation as POMDPs. The characterization enables tractable study of the problem and informs our method's design.

2. We develop a benchmark suite of asynchronous MuJoCo tasks for evaluation of RL in asynchronous regimes. The framework is general and can be readily applied to define AMDPs in new environments, facilitating future research and development.

3. We introduce **LANS**, a training strategy that regularizes policies by exposing them to a spectrum of noise regimes, leveraging *time-based* noising processes. We share both theoretical and empirical evidence to support our claims.

## 2 ASYNCHRONOUS ENVIRONMENTS

We establish the framework for RL in asynchronous environments, assuming familiarity with MDPs and POMDPs. We refer readers to Feinberg & Shwartz (2012); Krishnamurthy (2016) for comprehensive introductions. Our formalization proceeds in three steps: first, we define the concept of a *signal*; second, we extend MDPs to include asynchronicity; and third, we demonstrate the approximation of asynchronous MDPs as POMDPs.

Throughout this paper, we denote MDPs as tuples $(S, A, P_a, R_a)$, where $S$ and $A$ represent the state and action spaces, while $P_a$ and $R_a$ define the transition probabilities and reward functions conditioned on action $a \in A$. POMDPs also include an observation space $\Omega$ and conditional observation probabilities $O(\cdot \mid s \in S)$ that govern how states $s \in S$ result in partial observations $\Omega \ni o \sim O(o \mid s \in S)$.

## 2.1 Asynchronous Markov decision processes

In real-world applications, particularly in physical systems, observations often arrive from multiple sensors, providing the state space with a distinctive structure.

**Definition 1.** A **signaled state space** is a space $S$ together with subspaces $S_1, \ldots, S_n$, called **channels**, such that $S = \bigoplus_{i \leq n} S_i$.

Each channel $S_i$ corresponds to a distinct signal source (e.g., camera, thermometer, accelerometer) that reads information on the environment. In MDPs, we assume agents observe to the complete state $s_t \in S$ at each timestep $t \in \mathbb{N}$. The assumption fails in asynchronous environments, as sensors update at varying and possibly non-deterministic rates.

To represent missing signals, we extend each signal space $S_i$ with a null element denoted by $\perp$ that indicates an unreceived signal. We define $S_i^* = S_i \cup \perp$ as the extended signal space for channel $i$.

**Definition 2.** An **asynchronous Markov decision process** (**AMDP**) extends an MDP $\mathcal{M} = (S, A, P_a, R_a)$ with a signaled space $S = S_1 \times \cdots \times S_n$ by introducing:

- An observation space $\Omega = (\prod_{i=1}^{n} S_i^*) \times \mathbb{N}^n \times A^k$, where an observation $o \in \Omega$ comprises:
  - A vector of potentially null signals $(s_1^*, \ldots, s_n^*)$, where $s_i^* \in S_i^*$.
  - A vector of timestamps $(\delta^1, \ldots, \delta^n)$ indicating the elapsed time since each sensor's last reading.
  - A history of the $k$ most recent actions $(a_{t-k}, \ldots, a_{t-1})$.

- A signal renewal process for each channel $i$, described by renewal probabilities $p_i : \mathbb{N} \to [0, 1]$. Each value $p_i(\delta^i)$ is the probability that the $i$-th sensor provides a new reading after $\delta^i$ time steps since its last update.

The **observation component** for the $i$-th sensor at time $t \in \mathbb{N}$ is the vector $o_{(t,i)} = \left( s_{(t,i)}^*, \delta_t^i \right)$, which includes its value and time elapsed since its last non-null reading.

In an AMDP, the dynamics of the environment are still governed by the transition probabilities $P_a$, but the agent's observations are subject to asynchronous updates through the channel-specific renewal processes. For each channel $i$, the probability of receiving a new signal at time $t$ depends on $\delta_t^i$, the time elapsed since its last reading.

In an AMDP, we assume that the transition dynamics $\tilde{P}_a$ in the observation space $\Omega$ are separable by channel:

$$\tilde{P}_{a_t}(o_{t+1} \mid o_t) = \prod_{i=1}^{n} \tilde{P}_{(a_t,i)}(o_{(t+1,i)} \mid o_{(t,i)}). \tag{1}$$

In the following, we simplify the notation by removing the channel index $i$ unless necessary, and instead assume the existence of only one signal. Equation 1 guarantees that future developments about individual signals readily apply to general AMDPs.

The dynamics of signal updates are the described by their renewal process through the following equation:

$$\tilde{P}_{a_t}(o_{t+1} \mid o_t) = \begin{cases} p(\delta_t + 1) \cdot P_{a_t}(s_{t+1} \mid s_t) & \text{if } o_{t+1} = (s_{t+1}, 0) \\ 1 - p(\delta_t + 1) & \text{if } o_{t+1} = (\perp, \delta_t + 1) \\ 0 & \text{otherwise.} \end{cases} \tag{2}$$

At each time step $t \in \mathbb{N}$, a new reading $s_{t+1}$ is recorded with probability $p(\delta_t + 1)$ and the elapsed time value $\delta_{t+1}$ is reset. Alternatively, a null signal is received and the elapsed time value is increased by one $\delta_t + 1$.

## 2.2 Relationship between AMDPs and POMDPs

AMDPs are not a special case of POMDPs. To relate the two, one must define conditional probabilities $O(o \mid s)$ that describe the distribution of observations $o \in \Omega$ w.r.t. to the complete state $s \in S$.

The obstacle is the time-dependent dynamics of signal renewals. In an AMDP, the elapsed time value $\delta$ is necessary to derive the observation probabilities, the complete state alone is not sufficient. Instead, we state an approximation that treats AMDPs as POMDPs in the asymptotic regime.

**Proposition 1.** *Let $\mathcal{A}$ be an AMDP with stationary signal renewal process and well-defined rate $\lambda = \frac{1}{\mu}$. For $t \to \infty$, the process $\mathcal{A}$ can be approximated by a POMDP with conditional observation probabilities:*

$$O\left(o_t \mid s_t\right) \overset{t \to \infty}{\Rightarrow} \begin{cases} \lambda & \text{if } o = s \\ 1 - \lambda & \text{if } o = \bot. \end{cases} \tag{3}$$

*Proof.* Appendix A.1, leveraging the elementary renewal theorem (Ross, 2010). $\square$

For sufficiently long-running processes, the renewal probability becomes stationary and equals the inverse of the average inter-arrival period $\lambda$. An AMDP is then characterized by the vector $\boldsymbol{\lambda} = (\lambda_i)_{i \leq n} \in \mathbb{R}^n$. We use the notation $\mathcal{A}[\boldsymbol{\lambda}]$ to denote the POMDP approximating an AMDP $\mathcal{A}$ with converging sensor rates $\lambda_1, \ldots, \lambda_n$.

## 3 LANS

**Learning Across the Noise Spectrum** (**LANS**) is an off-policy training strategy for RL agents operating in asynchronous environments. It relies on the insight that we can view AMDPs as instances of a parameterized family of POMDPs, in which a parameter $\omega \in [0, 1]$ controls the degree of observability. By training across differentiated noise regimes—from highly impaired (low $\omega$) to the maximum available in the target environment ($\tilde{\omega}$)—we encourage the development of more robust policies. Algorithm 1 details LANS implementation.

Remarkably, LANS is not restricted to AMDPs and is virtually applicable to any class of parametrized POMDPs. The approach exploits the ability to simulate processes with higher noise ($\omega \leq \tilde{\omega}$) than that of the target environment. This requires knowledge of the noising process. In asynchronous environments, it can be simulated by appropriately masking observations, making AMDPs an ideal test case. Exploration of this idea in other categories of POMDPs is beyond the scope of our work, but we believe it is an exciting direction for future research.

### 3.1 NOISE PARAMETERIZATION OF AMDPS

In Section 2.2, we prove that an AMDP $\mathcal{A}$ can be approximated as a POMDP $\mathcal{A}[\boldsymbol{\lambda}]$ with stationary signal rates $\boldsymbol{\lambda} = (\lambda_i)_{i \leq n}$. The vector $\boldsymbol{\lambda}$ is a measure of the environment's stochasticity. A more compact representation is given by the **expected ratio of received signals**, the noise parameter:

$$\omega = \frac{1}{n} \sum_{i=1}^{n} \lambda_i. \tag{4}$$

The case $\omega = 1$ corresponds to the fully observable environment. As $\omega \to 0$, renewals become increasingly rare.

We employ an abuse of notation and use $\mathcal{A}[\omega]$ to denote the POMDP approximation of $\mathcal{A}$ with noise parameter $\omega$. Formally, $\mathcal{A}[\omega]$ comprises a collection of POMDPs, one for each $\boldsymbol{\lambda}$ that satisfies Equation 4. In practical implementations, given environment rates $\tilde{\boldsymbol{\lambda}} = (\tilde{\lambda}_i)_{i \leq n}$, we define $\mathcal{A}[\omega]$ by scaling $\tilde{\boldsymbol{\lambda}}$ by a factor of $c \in [0, 1]$:

$$\mathcal{A}[\omega] = \mathcal{A}\left[c\tilde{\boldsymbol{\lambda}}\right] \text{ with } \frac{1}{n} \sum_{i \leq n} c\tilde{\lambda}_i = \omega. \tag{5}$$

The map $\omega \mapsto \mathcal{A}[\omega]$ describes a family of noise-parametrized POMDPs, in which the target environment is realized at $\mathcal{A}[\tilde{\omega}]$ (corresponding to $c = 1$). The parametrization is smooth, see Appendix A.2 for details.

---

**Algorithm 1 LANS: Learning Across the Noise Spectrum**

---

**Input:** Parametric networks $\theta, \phi_1, \phi_2$ for actor and Q-value functions.
  $c \in (0, 1)$ defining minimum noise simulation ($\omega_{\min} = c \cdot \tilde{\omega}$).
  $K, B, L$: optimization steps per rollout, batch size, and trajectory length.
  $R, T$: number of rollouts, max rollout length.
  **for** each rollout $r = 1 \ldots R$ **do**
    Collect a rollout $\tau \sim \mathcal{A}[\tilde{\omega}]$ with policy $a_t \sim \pi(\cdot \mid \mu_\theta, \sigma_\theta)$
    $\mathcal{D} \leftarrow \mathcal{D} \cup \{\tau_t\}_{t \leq T}$
    **for** $K$ steps **do**
      $D \leftarrow (\tau^b_{[l_b:l_b+L]}) \sim \mathcal{D}$
      NOISESIMULATION$(D, c)$                                      ▷ Core LANS mechanism
      **if** $s^*_{(t,i)} = \bot$ **then** $s^*_{(t,i)} \leftarrow s^*_{(t-\delta^i_t, i)}$ **end if**     ▷ Do this across all signals
    **end for**
    UPDATECRITIC$(D)$
    UPDATEACTOR$(D)$
  **end for**
  **procedure** NOISESIMULATION$(D, c)$
    $(c_b)_{b \leq B} \sim \mathcal{U}_{[c,1]}$                       ▷ Each $c_b$ is drawn randomly from a uniform distribution
    **for** each trajectory $\tau^b$ in $D$ **do**                     ▷ Can be performed concurrently
      **if** Bernoulli$(c_b) = 1$ **then** $s^*_{(t,i)} \leftarrow \bot$ **end if**     ▷ Do this across all signals
    **end for**
  **end procedure**

---

## 3.2 METHOD

LANS extends the training domain of an RL algorithm from the target AMDP $\mathcal{A}[\tilde{\omega}]$ to a collection of asynchronous environments $\mathcal{A}[\omega]$ drawn from a noise range $\omega \in [\omega_{\min}, \tilde{\omega}]$.

**Algorithm and architecture** We use SAC (Haarnoja et al., 2018) with two Q-value networks (Dankwa & Zheng, 2020) as the actor-critic loss. We employ RNNs for the architectural backbone of our models, particularly gated recurrent units (GRU) (Bahdanau et al., 2014). The actor $\theta$ and Q-value networks $\phi_1, \phi_2$ use separate RNNs and do not share weights. Mathematically, $\phi_i\left(o_t, a_t, h_t^{\phi_i}\right) = \left(Q_{\phi_i}\left(o_t, a_t, h_t^{\phi_i}\right), h_{t+1}^{\phi_i}\right)$ and $\theta\left(o_t, h_t^\theta\right) = \left(\mu_t, \sigma_t, h_{t+1}^\theta\right)$, i.e. the three functions take the last hidden state $h_t$ as one of their inputs and compute the next hidden state $h_{t+1}$ together with Q-values and policy distribution parameters.

**Data** Training alternates between data collection and optimization (Williams, 1992). During collection, we sample a full rollout $\tau = (o_t, a_t, r_t)_{t \leq T}$ by running the SAC stochastic policy on the target environment $\mathcal{A}[\tilde{\omega}]$ and store it on a replay buffer $\mathcal{D}$ (Lin, 1992). After each sampled rollout, we perform $K$ optimization steps on batches drawn from $\mathcal{D}$. A batch $D = \left(\tau^b_{[l_b:l_b+L]}\right) \in \mathbb{R}^{B \times L \times (\dim O + \dim A + 1)}$ is a tensor storing observations, actions, and rewards for $B$ trajectories of length $L$, padded if necessary.

**Noise simulation** Rollouts are drawn from the target environment, yielding batches $D \sim \mathcal{A}[\tilde{\omega}]$. Before each optimization step, we simulate noisier samples $D^* \sim \mathcal{A}[\omega_{\min} : \tilde{\omega}]$ by uniformly drawing $\omega_b \in [\omega_{\min}, \tilde{\omega}]$ for each trajectory $\tau^b$ and randomly masking a percentage $\frac{\tilde{\omega} - \omega_b}{\tilde{\omega}}$ of signal readings:

$$s^*_t \mapsto \begin{cases} \bot & \text{if Bernoulli}\left(\frac{\tilde{\omega} - \omega_b}{\tilde{\omega}}\right) = 1 \text{ or } s^*_t = \bot \\ s^*_t & \text{otherwise.} \end{cases} \tag{6}$$

The elapsed time values $\delta_t$ are also updated accordingly. In Algorithm 1, this step corresponds to the NOISESIMULATION procedure.

As of Definition 2, observation components $o_t = \left[s^*_t, \delta_t, a_{[t-k:t]}\right]$ include a history $a_{[t-k:t]}$ of the $k$ most recent actions and times $\delta$ elapsed since each last renewal. Before being provided as inputs to

the models, we replace each unreceived signal $s_t^* = \bot$ with its most recently observed value $s_{t-\delta_t}^*$. The decision provides more meaningful inputs to the networks and is crucial for the developments of Section 3.3.

**Hyperparameters** *Rollout and data* parameters include the rollout horizon $T$, the number of optimization steps per rollout $K$, batch size $B$ (measured in number of sub-trajectories), and the sub-trajectory length $L$. *Noise* parameters comprise the minimum noise factor $c$, which defines the lower bound $\omega_{\min} = c \cdot \tilde{\omega}$ of the simulated noise spectrum. *Optimization* parameters include the optimizer's learning rate and the entropy regularization coefficient in SAC. Appendix B reports the values used for our experiments.

## 3.3 LANS AND REGULARIZATION

LANS encourages learning invariant representations of observations with respect to time-dependent noising, which implies more robust modeling (Zhang et al., 2021a). Here, we elaborate on the regularization mechanism rigorously and prove our core result.

RL algorithms often learn parametric policies $\pi_\theta (a \mid o) = \mathcal{P}_\rho (a, \rho = \theta (o))$ (Bishop, 2006), which are functions $\theta : \Omega \to \mathbb{R}^m$ outputting the parameters $\rho$ of a finite-dimensional distribution $\mathcal{P}_\rho$ on $A$. Therefore, we assume $\pi : \Omega \to \mathbb{R}^m$ to be a deterministic function.

Given a sequence of observations $(o_t)_{t \leq T}$, the NOISESIMULATION procedure of Algorithm 1 applies a random transformation $G : (o_t)_{t \leq T} \mapsto (o_t^\star)_{t \leq T}$ *projecting* the sequence into a lower dimensional subspace. LANS can be interpreted as implicitly minimizing the following loss function:

$$\mathcal{L}_{\text{LANS}} (o) = \mathbb{V}_{o^\star \sim G(o)} (\pi (o^\star)). \tag{7}$$

Policy predictions must have contained variance w.r.t the random transformations $G$ that project observation sequences into less informative subspaces.

In the primary result of our work, we prove that LANS acts a regularization term:

**Proposition 2.** *If the policy $\pi$ is $C^1$, up to second order in $(o^\star - o)$,*

$$\mathcal{L}_{\text{LANS}} (\pi) = \mathbb{E}_{o \sim \mathcal{D}} \left[ \text{tr} \left( J_\pi (o) \Sigma_G (o) J_\pi (o)^\top \right) \right] + o \left( \| o^\star - o \|^2 \right) \tag{8a}$$

$$\Sigma_G (o) \approx \text{diag} \left( p (1 - p) \Delta_i (o)^2 \right) \text{ with } \Delta_i (o) = o_i^\star - o_i. \tag{8b}$$

*Proof.* Proof provided in Appendix A.3. $\square$

Expanding Equation 8a, we obtain:

$$\mathcal{L}_{\text{LANS}} (\pi) \approx \mathbb{E}_{o \sim \mathcal{D}} \left[ \sum_i p (1 - p) \Delta_i (o) \| \partial_{o_i} \pi (o) \|_2^2 \right]. \tag{9}$$

The loss $\mathcal{L}_{\text{LANS}}$ acts as a minimization term on the policy's derivatives—i.e., a Jacobian penalty (Rifai et al., 2011; Sokolić et al., 2017; Novak et al., 2018). The effect is proportional on the value of $\Delta_i (o)$, which quantifies the shift between two consecutive readings. The regularization penalty is *strongest in the directions most sensitive to the dynamics of renewal*.

## 4 RELATED WORK

**Partially observable Markov decision processes** Extensive research addresses POMDPs in RL. It includes learning the dynamics of systems from individual frames (Hausknecht & Stone, 2015; Payne et al., 2024), path-planning (Xie et al., 2021), and multi-agent cooperation (Oroojlooyja-did & Hajinezhad, 2019; Papoudakis et al., 2020). Other works leverage POMDPs to investigate traditional RL settings such as Meta-RL (Schmidhuber, 1987; Soorki et al., 2023; Parisotto et al., 2020) and generalization performance (Rajeswaran et al., 2017; Agarwal et al., 2020). Memory architectures, especially RNNs, have quickly become popular as the standard approach in partially

observable environments (Schmidhuber, 1990; Lu et al., 2023). Model-based approaches learn to recover ground-truths states from partial observations (Han et al., 2020; Ren et al., 2023; Lee et al., 2020). On the opposite, model-free architectures have objectives that focus solely on reward maximization (Meng et al., 2021; Mirowski et al., 2017). Ni et al. (2022) demonstrate that model-free approaches are sufficient to tackle POMDPs in most scenarios, provided careful design of the architectural backbone.

**Noise-based regularization**  Regularization (Hastie et al., 2009) aims at constraining the complexity of machine learning models with positive effects on robustness (Jeong & Shin, 2020; Li & Zhang, 2021) and generalization (Loshchilov & Hutter, 2017; Wan et al., 2013; Li et al., 2018) (but see Zhang et al. (2017; 2021b) for counterarguments). Among regularization techniques, noise-based (Sajjadi et al., 2016) ones work by introducing noise on the inputs (An, 1996; Bishop, 1995), weights (Fortunato et al., 2018; Rakin et al., 2018), or outputs (Zhang & Sabuncu, 2018). Dropout (Srivastava et al., 2014; Gal & Ghahramani, 2015) is a popular regularization strategy that relies on random masking of intermediate network's reprsentations. In Section 3.3, we show that LANS works as a noise-based regularization approach. Denoising auto-encoders (DAE) (Vincent et al., 2008) are especially relevant to our work, due to their noise-injection mechanism based on masking of random coordinates. In the experiments, we show that for AMDPs, LANS is a more effective strategy than DAEs.

**Data augmentation**  LANS' core mechanism is the NOISESIMULATION procedure of Algorithm 1, which applies transformations to the input data before performing a gradient step. Data augmentation (Shorten & Khoshgoftaar, 2019; Park et al., 2019) refers to the class of strategies that exploit symmetries (Chen et al., 2020; Bjerrum, 2017) in the data distribution to simulate larger datasets. Data augmentation is prevalent in RL (Sun et al., 2024), with most works focusing on visual observations (Yarats et al., 2021b;a; Zhang et al., 2021a). Despite the similarity, data augmentation differs from LANS because of its focus on transformations that preserve the semantics of samples (Trabucco et al., 2024). Noising is occasionally treated as a data augmentation strategy (Iglesias et al., 2023); Xie et al. (2020) discuss the advantages in deploying realistic noising processes, which is what LANS achieves w.r.t AMDPs.

## 5 EXPERIMENTS

We share a framework and benchamrk suite for AMDPs definition, and we conduct experiments on standard RL environments modified to incorporate asynchronicity. We compare LANS against alternative regularization techniques—DAEs and Gaussian additive noise—and a no-regularization baseline, proving that our strategy (i) improves learning performance and/or variance in asynchronous environments and (ii) it is more effective than other forms of regularization.

### 5.1 ASYNCHRONOUS RL AND MUJOCO ENVIRONMENTS

We develop a dedicated codebase for asynchronous RL environments. The library builds on top of PyTorch (Ansel et al., 2024), Gymnasium (Towers et al., 2024), and TorchRL (Bou et al., 2023), and provides a modular implementation of AMDPs. It allows users to specify processes for signals, simulate asynchronicity, and interface with RL pipelines. We include detailed documentation to facilitate reproducibility and adoption.

For evaluation, we adapt the widely used MuJoCo continuous control benchmarks Todorov et al. (2012) from the DeepMind Control Suite (Tassa et al., 2018). In our formulation, each joint state—comprising position, velocity, and rotational coordinates—is modeled as a signal. Asynchronicity is introduced by assigning each channel a renewal process. Concretely, we sample renewal intervals from a Gamma distribution $\delta^i \sim \text{Gamma}(1, 1)$, truncated to $\delta \leq 4$ steps. This results in an average signal ratio of $\tilde{\omega} \approx 0.39$, i.e., agents observe $\sim 40\%$ of the signals, on average.

We construct a suite of asynchronous MuJoCo benchmarks—including *Async-HalfCheetah*, *Async-Hopper*, *Async-Ant*, *Async-Walker2d*, and *Async-Reacher*. By grounding our experiments in these standard tasks (Duan et al., 2016), we retain comparability with prior work while introducing a controlled source of asynchronicity that stresses the ability of algorithms to operate on AMDPs.

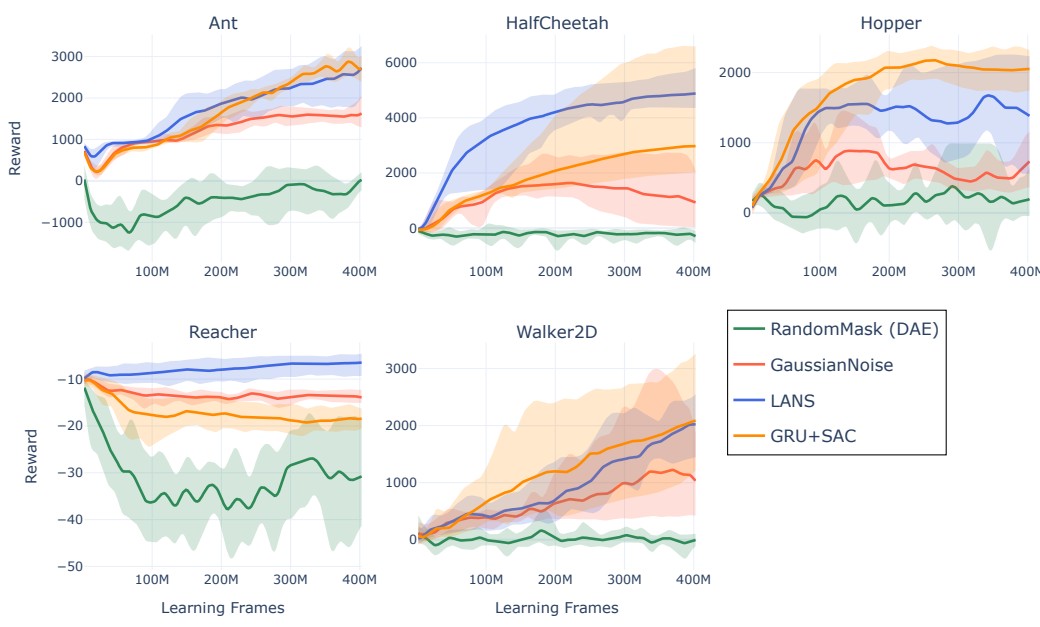

Figure 2: Training curves for LANS, GRU+SAC, DAE, and GAUSSNOISE, averaged across five runs per environment. For all environments except Async-Hopper, LANS either outperforms other methods after $400M$ learning frames or exhibits more controlled variance.

Table 1: Average reward per episode after $400M$ training frames, averaged across 5 runs per task, variance in parentheses. Bold values indicate best within the environment. LANS achieves either best returns on Async-HalfCheetah and Async-Reacher, or controlled variance for other environments, except Async-Hopper. Alternative regularization algorithms do not display competitive results.

| ALGORITHM | Ant (K) | HalfCheetah (K) | Hopper (K) | Walker2D (K) | Reacher |
|---|---|---|---|---|---|
| LANS | 2.6 (0.8) | **4.9** (0.6) | 1.4 (0.8) | **2.0** (0.5) | **−6.7** (2.9) |
| GRU+SAC | **2.8** (0.6) | 3.0 (2.1) | **2.1** (0.2) | **2.0** (0.9) | −18 (2.2) |
| DAE | 0.0 (0.3) | −0.2 (0.05) | 0.2 (0.4) | 0.0 (0.1) | 0.0 (0.1) |
| GAUSSNOISE | 1.6 (0.5) | 1.0 (0.6) | 0.7 (0.5) | 1.1 (0.4) | 1.1 (0.4) |

## 5.2 ANALYSIS ACROSS ENVIRONMENTS

We compare LANS with three baselines across five environments, while retaining the same architectural backbone: a SAC loss function with GRU networks modeling actor and critic functions. **GRU+SAC** is the standard baseline, trained without modifications to the original algorithm. Each experiment is run five times to control for random oscillations and to estimate variance.

**LANS** Our approach, detailed in Algorithm 1. Compared to the standard baseline, it requires setting the additional hyperparameter $c \in [0, 1]$. We choose $c = 0.5$ for all experiments except Hopper, where a value of $c = 0.1$ shows minor performance improvements.

**DAE** We borrow a technique from denoising autoencoder (Vincent et al., 2008), consisting of training the network while masking a fixed percentage $\nu \in [0, 1]$ of input coordinates at every gradient step. The decision is informed by the method's similarity to LANS: it is obtained by replacing the NOISESIMULATION procedure in Algorithm 1 with one that masks input coordinates in a *time-unaware* fashion. We adopt $\nu = 0.5$ for all our experiments.

**GAUSSNOISE**   Gaussian additive noise applied to inputs underlies an established regularization technique (Bishop, 1995). GAUSSNOISE modifies the NOISESIMULATION procedure of Algorithm 1, replacing random signal masking with Gaussian shifts.

Figure 2 shows training curves for LANS and other algorithms, displaying the evolution of average episode return per frame. Table 1 reports average episode return of each model after $400M$ learning frames. In most tasks, LANS achieves either the highest returns or the most controlled variance. In the case of Async-HalfCheetah, the performance is better for a factor of $> 1.5\times$ than the one of the standard baseline. Async-Hopper is the only environment in which LANS' underperfoms; it is the easiest task among the five and therefore might not provide sufficient complexity to detect regularization impact. For Async-Ant and Async-Walker2d, there are no visible performance gains, but significantly lower variance. Across all experiments, competing regularization algorithms fail to learn meaningful policies, demonstrating their differences with LANS to be crucial.

## 5.3   CURVATURE ANALYSIS

In Section 3.3, we establish LANS as a regularization mechanism. To validate this claim, we estimate the curvature of policies and compare them with GRU+SAC. We approximate the Hessian of the policy output with respect to observations using finite differences, and report its Frobenius norm as a curvature measure. Estimates are computed over $1K$ observations sampled from the evaluation environment.

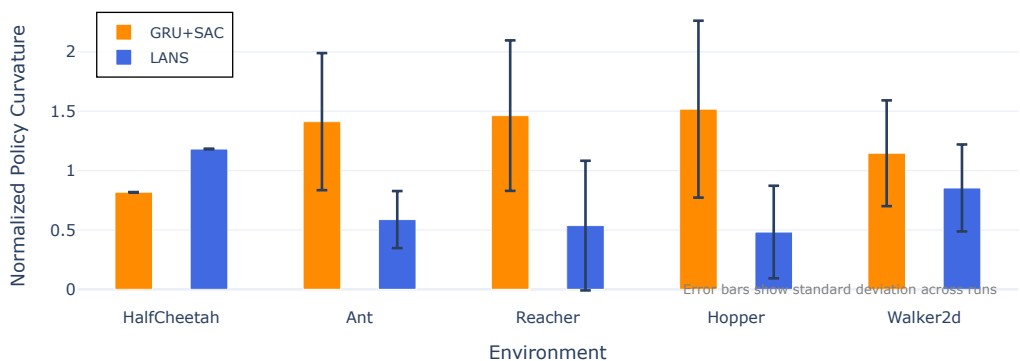

Figure 3: Estimated policy curvature across tasks. Bars show the Frobenius norm of the Hessian of the learned policy, averaged over $1,000$ observations and five random seeds, and normalized. Error bars denote standard deviation across seeds. LANS produces policies with lower curvature, except for Async-HalfCheetah.

Figure 3 shows the average curvature values for policies trained with and without LANS, which have been normalized to preserve scale. Results are aggregated across five random seeds, with error bars representing standard deviation. With the exception of Async-HalfCheetah, LANS yields policies with lower curvature, sometimes by a factor of $< 0.5$. By exposing agents to a spectrum of noise regimes, LANS acts as a regularizer that penalizes excessive curvature of the policy function.

## 6   CONCLUSION

We introduce **Learning Across the Noise Spectrum (LANS)**, an approach to reinforcement learning in asynchronous environments. We formalize asynchronous environments as noise-parameterized POMDPs, and develop a theoretical foundation linking LANS to regularization. We demonstrate performance improvements across modified MuJoCo benchmarks. Our research establishes LANS as an effective solution for real-world robotic systems operating with imperfect sensory information, bridging a gap between theoretical RL frameworks and practical deployment in industrial settings.

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

# A  PROOFS

In this appendix, we provide proofs to the propositions and claims contained in the main paper. Throughout this section, we restrict the discussion to the case where an AMDP is described by only one signal. This way, we omit the index $i$ (e.g. $o_{(t,i)} = \left( s^*_{(t,i)}, \delta^i_t \right)$ becomes $o_t = (s^*_t, \delta_t)$) referring to channels in our notation, easing readability. Equation 1 ensures that the multi-signals case can be reduced to the single one.

## A.1  APPROXIMATION OF AMDPs AS POMDPs

We prove Proposition 1, stating that AMPDs—as defined in Definition 2—can be approximated by POMDPs in the asymptotic regime. The proof relies on the theory of renewal processes and especially on the Erdös-Feller-Pollard theorem. Before proceeding, we provide the essential definitions and assumptions that underlie the next developments. We refer to Mitov & Omey (2014) for a source on the material exposed here.

**Definition 3** (Renewal Process). Let $X_n \in \mathbb{N}^+$ be a sequence of i.i.d. positive random variables. The increasing sequence of partial sums variables $S_n$ defined as

$$S_n = \sum_{i=0}^{n} X_i \tag{10}$$

is called a **renewal process** and the $S_n$ **renewal times**.

A renewal process describes the dynamics of events that occur at possibly non-deterministic frequency, with independent and identically distributed renewal intervals.

Crucially, renewal processes describe the asynchronicity of signals in an AMDP. Given a sequence of observations $(o_t = (s^*_t, \delta_t))_{t>0}$, we can restrict to the sub-sequence of observed readings $(o_{t_n})_{n \geq 0}$ such that $\delta_{t_n} = 0$ and $\delta_u \neq 0, \forall u \neq t_n$. The sequence $\Delta_n = t_{n+1} - t_n$ is that of the renewal intervals between readings. The variables $\Delta_n$ are i.i.d. as a consequence of Equation 2:

$$P(\Delta_n = k) = p(k+1) \sum_{j=1}^{k} (1 - p(j)). \tag{11}$$

In particular, $P(\Delta_n = k)$ does not depend on $n$ nor the value of $\Delta_m$ for $m < n$. Therefore, the sequence of reading times $\Sigma_n = \sum_{i=1}^{n} \Delta_i$ is a renewal process.

Let $r_t = P(\delta_t = 0)$ denote the probability that there is an observed signal at time $t \geq 0$. We have

$$r_t = \sum_{n=1}^{\infty} P(\Sigma_n = t). \tag{12}$$

To prove Proposition 1, we must prove that $\lim_{t \to \infty} r_t = \lambda$ where $\lambda = \frac{1}{\mathbb{E}[\Delta_n]}$ indicates the signal renewal rate, which is the reciprocal of the average renewal time.

The Erdös-Feller-Pollard theorem directly proves Equation 12, but it relies on two assumptions about the renewal process induced by $\Delta_n$.

**Assumption 1.** *The average inter-arrival renewal time $\mu = \mathbb{E}[\Delta_n]$ is finite.*

**Assumption 2.** *For each arithmetic progression $k\mathbb{N} = \{kn \mid n \in \mathbb{N}\}$ with $k \geq 2$, we have*

$$\sum_{i \in k\mathbb{Z}} P(\Delta_n = i) < 1. \tag{13}$$

The first assumption poses a probabilistic bound on renewal times. Assumption 2 states that the renewal process is **aperiodic**, meaning there is no $k \geq 2$ such that renewals only occur at times $t = k \cdot u \in k\mathbb{N}$.

We remark that for the purpose of our work, it is safe to assume that AMDPs satisfy Assumptions 1 and 2. Sensors with infinite average renewal times would provide unreliable information that can potentially be disrupted indefinitely. For sensors with renewal times concentrated on a periodic subsequence $t \in k\mathbb{N}$, we can express the problem in terms of observations drawn at time-steps in $k\mathbb{N}$ and Assumption 2 would hold.

Finally, we state Erdös-Feller-Pollard theorem.

**Theorem 1** (Erdös-Feller-Pollard). *Let $S_n = \sum_{i=1}^{n} X_n$ be a renewal process satisfying Assumptions 1 and 2 and let $\mu = \mathbb{E}[X_n]$. Then*

$$r_t = \sum_{n=1}^{\infty} P(S_n = t) \overset{t \to \infty}{\to} \frac{1}{\mu}. \tag{14}$$

Proposition 1 follows as a corollary.

**Proposition 3.** *Let $\mathcal{A}$ be an AMDP with stationary signal renewal processes and well-defined rates $\lambda_i = \frac{1}{\mu_i}$. The process $\mathcal{A}$ can be approximated by a POMDP with conditional observation probabilities:*

$$O(o \mid s) = \prod_{i=1}^{n} O_i(o_i \mid s_i) \tag{15}$$

*where for each channel $i$:*

$$O_i(o_i \mid s_i) = \begin{cases} \lambda_i & \text{if } o_i = s_i \\ 1 - \lambda_i & \text{if } o_i = \perp. \end{cases} \tag{16}$$

*Proof.* Since Equation 2 separates transition probabilities by signals, we will restrict our proof to the one-signal case, omitting indexes indicating channels. The full statement of the proposition is obtained by application across signals.

Equation 16 states that the variable $R_t = \mathbf{1}_{\{o_t = s_t\}} \sim \text{Bernoulli}(\lambda)$. For an AMPD, the variable $Q_t = \mathbf{1}_{\{\delta_t = 0\}} \sim \text{Bernoulli}(r_t)$. Theorem 1 implies $r_t \overset{t \to \infty}{\to} \lambda = \frac{1}{\mu}$. $\qquad\square$

### A.2 Smoothness of noise-parametrization of AMPDs

In Section 3.1, we state that the AMDPs $\mathcal{A}[\omega]$ form a parametrization of POMDPs that is **smooth** for $\omega \in [0, 1]$. We formalize and prove this notion.

The AMDPs $\mathcal{A}[\omega]$ are characterized by their conditional probabilities $O_\omega : \Omega \times S \to \mathbb{R}$, $(o, s) \overset{\omega}{\mapsto} O_\omega(o \mid s)$. The parametrization is smooth if the $O_. : [0, 1] \to (\Omega \times S \to \mathbb{R})$ is smooth w.r.t. $\omega \in [0, 1]$ in the Wasserstein space $\mathcal{P}_1(\Omega \times S)$ of probability measures Ambrosio et al. (2008).

**Definition 4** (Wassertein metric). Let $(M, d)$ be a metric space and $P, Q$ two probability distributions on $M$ with finite expected value. The **Wassertein distance** $W_1(P, Q)$ between $P$ and $Q$ is defined as

$$W_1(P, Q) = \inf_{\gamma \in \Gamma(P, Q)} \mathbb{E}_{(x,y) \sim \gamma}[d(x, y)] \tag{17}$$

where $\Gamma(P, Q)$ denotes the set of **couplings** between $P$ and $Q$, i.e. a coupling $\gamma$ is a distribution whose marginal probabilities equal $P$ and $Q$ respectively. The metric space $(\mathcal{P}_1(M), W_1)$ is called the **Wassertein space**.

Before proceeding to verify the smoothness of $\mathcal{A}[\omega]$, we must specify the structure of a metric space for the observation space $\Omega$. While in general $S \subseteq \mathbb{R}^m$ for some $m \in \mathbb{N}$, the existence of $\perp \in \Omega$ implies that $\Omega$ does not naturally embed into a real space. In practice, however, we can choose $\perp = \mathbf{0}$ and equate the lack of a signal with the null vector. More than just a theoretical trick, this choice describes the concrete implementation of LANS as described in Section 3.2, since unobserved

signals are replaced with their most recent reading, or masked to the null vector if no most recent observation is available.

**Proposition 4.** *For each* $\omega \in [0, 1]$

$$\lim_{h \to 0} \frac{W_1\left(O_{\omega+h}\left(\cdot \mid s\right), O_\omega\left(\cdot \mid s\right)\right)}{|h|} = d\left(s, \perp\right). \tag{18}$$

*Proof.* We observe that $O_{\omega+h}\left(\cdot \mid s\right)$ and $O_\omega\left(\cdot \mid s\right)$ have both support on the two-elements set $\{s, \perp\}$. Therefore, any coupling $\gamma_{\omega,h}\left(s\right)$ is identified by a probability matrix

$$\gamma_{\omega,h}\left(s\right) \sim \begin{pmatrix} B_s = P\left(s, s\right) & F_s = P\left(s, \perp\right) \\ F_\perp = P\left(\perp, s\right) & B_\perp = P\left(\perp, \perp\right) \end{pmatrix}. \tag{19}$$

We have

$$B_s + F_s = O_\omega\left(s \mid s\right) \tag{20a}$$
$$B_s + F_\perp = O_{\omega+h}\left(s \mid s\right) \tag{20b}$$
$$F_s + F_\perp = P\left(O_\omega\left(\cdot \mid s\right) \neq O_{\omega+h}\left(\cdot \mid s\right)\right). \tag{20c}$$

From Equations 20a and 20b it follows

$$|h| = |O_\omega\left(s \mid s\right) - O_{\omega+h}\left(s \mid s\right)| = \tag{21a}$$
$$= |B_s + F_s - B_s - F_\perp| = \tag{21b}$$
$$= |F_s - F_\perp|. \tag{21c}$$

Because $F_s$ and $F_\perp$ are both non-negative, it must hold $F_s + F_\perp \geq |h|$.

We now observe

$$\mathbb{E}_{(x,y)\sim\gamma_{\omega,h}(s)}\left[d\left(x, y\right)\right] = P_{(x,y)\sim\gamma_{\omega,h}(s)}\left(x \neq y\right) d\left(s, \perp\right) = \tag{22a}$$
$$= \left(F_s + F_\perp\right) d\left(s, \perp\right) \geq \tag{22b}$$
$$\geq |h| d\left(s, \perp\right), \tag{22c}$$

which implies

$$W_1\left(O_{\omega+h}\left(\cdot \mid s\right), O_\omega\left(\cdot \mid s\right)\right) \geq |h| d\left(s, \perp\right). \tag{23}$$

For $h > 0$, a possible coupling $\tilde{\gamma}_{\omega,h}\left(s\right)$ is given by

$$\tilde{\gamma}_{\omega,h}\left(s\right) = \begin{pmatrix} \omega & 0 \\ |h| & 1 - \omega - h \end{pmatrix}, \tag{24}$$

and for $h < 0$

$$\tilde{\gamma}_{\omega,h}\left(s\right) = \begin{pmatrix} \omega + h & |h| \\ 0 & 1 - \omega \end{pmatrix}, \tag{25}$$

which together imply

$$\mathbb{E}_{(x,y)\sim\tilde{\gamma}_{\omega,h}(s)}\left[d\left(x, y\right)\right] = |h| d\left(s, \perp\right) \implies \tag{26a}$$
$$\implies W_1\left(O_{\omega+h}\left(\cdot \mid s\right), O_\omega\left(\cdot \mid s\right)\right) \leq |h| d\left(s, \perp\right). \tag{26b}$$

Combining Equations 23 and 26b we obtain

$$W_1\left(O_{\omega+h}\left(\cdot \mid s\right), O_\omega\left(\cdot \mid s\right)\right) = |h| d\left(s, \perp\right) \tag{27}$$

and finally

$$\lim_{h \to 0} \frac{W_1\left(O_{\omega+h}\left(\cdot \mid s\right), O_\omega\left(\cdot \mid s\right)\right)}{|h|} = \lim_{h \to 0} \frac{|h|}{|h|} d\left(s, \perp\right) = d\left(s, \perp\right). \tag{28}$$

$\square$

### A.3 LANS AND REGULARIZATION

We continue the discussion in Section 3.3 and prove Proposition 2. We recall the implicit LANS loss at an input $o \in \Omega$:

$$\mathcal{L}_{\text{LANS}}(o) \doteq \text{Var}_{o^\star \sim G(o)}\big(\pi(o^\star)\big) = \mathbb{E}_{o^\star \sim G(o)}\left[\big\|\pi(o^\star) - \mathbb{E}_{u^\star \sim G(o)}[\pi(u^\star)]\big\|_2^2\right]. \tag{29}$$

**Proposition 5.** *Suppose $\pi$ is continuously differentiable in a neighborhood of $o$, and let $\Delta \doteq o^\star - o$. Then, up to second order in $\Delta$,*

$$\mathcal{L}_{\text{LANS}}(o) = \text{tr}\big(J_\pi(o)\,\Sigma_G(o)\,J_\pi(o)^\top\big) + o(\|\Delta\|^2), \tag{30}$$

*where $J_\pi(o) \in \mathbb{R}^{m \times d}$ is the Jacobian of $\pi$ at $o$, and $\Sigma_G(o) = \text{Cov}(o^\star - o \mid o)$ is the covariance of the perturbation induced by $G$.*

*Proof.* By a first-order Taylor expansion of $\pi$ around $o$ we have

$$\pi(o^\star) = \pi(o) + J_\pi(o)\,\Delta + R(\Delta), \tag{31}$$

where the remainder $R(\Delta)$ satisfies $\|R(\Delta)\| = o(\|\Delta\|)$. Taking expectation under $G(o)$ gives

$$\mathbb{E}[\pi(o^\star) \mid o] = \pi(o) + J_\pi(o)\,\mathbb{E}[\Delta \mid o] + o(\|\Delta\|). \tag{32}$$

Subtracting this mean from the expansion yields

$$\pi(o^\star) - \mathbb{E}[\pi(o^\star) \mid o] = J_\pi(o)\,(\Delta - \mathbb{E}[\Delta \mid o]) + o(\|\Delta\|). \tag{33}$$

Therefore, to second order in $\Delta$,

$$\mathcal{L}_{\text{LANS}}(o) = \mathbb{E}\left[\big\|\pi(o^\star) - \mathbb{E}[\pi(o^\star) \mid o]\big\|_2^2\right] \tag{34}$$

$$= \mathbb{E}\big[\|J_\pi(o)\,(\Delta - \mathbb{E}[\Delta \mid o])\|_2^2\big] + o(\|\Delta\|^2) \tag{35}$$

$$= \text{tr}\big(J_\pi(o)\,\Sigma_G(o)\,J_\pi(o)^\top\big) + o(\|\Delta\|^2), \tag{36}$$

where the last equality follows since $\mathbb{E}[(\Delta - \mathbb{E}[\Delta])(\Delta - \mathbb{E}[\Delta])^\top] = \Sigma_G(o)$ by definition. $\square$

Under the conditions of Proposition 5,

$$\mathcal{L}_{\text{LANS}}(\pi) \doteq \mathbb{E}_{o \sim \mathcal{D}}\big[\mathcal{L}_{\text{LANS}}(o)\big] \approx \mathbb{E}_{o \sim \mathcal{D}}\big[\text{tr}\big(J_\pi(o)\,\Sigma_G(o)\,J_\pi(o)^\top\big)\big], \tag{37}$$

so training with LANS is equivalent to adding a Jacobian regularizer weighted by the covariance $\Sigma_G(o)$ of the temporal noise process.

**LANS** For the noising process used by LANS, each coordinate $i \in \{1, \ldots, n\}$ is replaced by its most recent past value $\tilde{o}_i$ with probability $p_i$, and kept unchanged with probability $1 - p_i$. The perturbation along dimension $i$ is therefore

$$\Delta_i = \begin{cases} \tilde{o}_i - o_i, & \text{with probability } p_i, \\ 0, & \text{with probability } 1 - p_i. \end{cases} \tag{38}$$

Assuming independence across coordinates, the covariance matrix $\Sigma_G(o)$ is diagonal with entries

$$[\Sigma_G(o)]_{ii} = p_i(1 - p_i)\,\Delta_i^2. \tag{39}$$

Plugging this expression into Proposition 5 yields

$$\mathcal{L}_{\text{LANS}}(o) \approx \sum_{i=1}^n p_i(1 - p_i)\,\Delta_i^2\,\|\partial_{o_i}\pi(o)\|_2^2. \tag{40}$$

## B HYPERPARAMETERS AND STATISTICS

### B.1 HYPERPARAMETERS

The hyperparameters most relevant to the various methods are reported in the main paper. In this section, we report architecture and loss function parameters omitted from the main work.

**Network**  We adopt the same hyperparameters relative to network size for every experiment. We use a depth of 1 and a hidden size of 128 both for the actor GRU and the Q-value one. Observations are embedded in a latent space of dimensionality 32 both before being provided to the RNN. They are embedded again with the same dimensionality, but different weights, and concatenated to the output of the RNN, which is then processed through an feed-forward network of depth 2 and hidden dimension 256. We adopt ReLU for nonlinearities.

**Loss function**  Rewards are scaled by a factor of 5 during training. We use an interpolation factor $\tau$ of $5 \times 10^{-3}$ and a discount factor $\gamma$ of $0.99$ for SAC. We rely on automatic entropy regularization with the same learning rate as the rest of the network, i.e. $3 \times 10^{-4}$.

**Training**  We train each run for $1,536$ rollouts, each with a maximum of $1,024$ frames. We adopt an initial random exploration phase for 16 rollouts. We perform 128 gradient step per trajectory. The batch size of each gradient step is $2,048$ frames, split across sub-trajectories of maximum length 64, implying a sequence batch size of 32.

**Evaluation**  We do not apply reward scaling during evaluation. We perform an evaluation cycle every $1,024$ gradient steps. We run each evaluation cycle by collecting the average total reward of an episodes across 16 instances.

## B.2 TRAINING TIMES

We run all our experiments on NVIDIA Tesla P100, on which a single run takes between 44 and 56 hours of processing time.

## C  USAGE OF LLMS

Large language models (LLMs) were employed in the preparation of this work exclusively for auxiliary purposes. They were used to refine the exposition by improving grammar, clarity, and stylistic consistency, but not for the generation of substantive scientific content or entire paragraphs. Additionally, LLMs were leveraged to automate repetitive aspects of code development, particularly in the production of scripts for generating plots and figures. All theoretical contributions, derivations, experimental design, and main textual content were authored by the researchers.

