# OpenReview forum: "Learning Across the Noise Spectrum: An Approach to Reinforcement Learning with Asynchronous Signals"
_ICLR.cc/2026/Conference — Submitted to ICLR 2026_

### Official Review · Reviewer_CuFk · 2025-11-01

**Soundness:** 2
**Presentation:** 2
**Contribution:** 2
**Rating:** 2
**Confidence:** 2

**Summary:**

The works studies reinforcement learning in asynchronous environments, where agents receive only a subset of all available signals at each timestep. The authors formalize asynchronous MDPs and connect them to a parametrized family of POMDPs. Then, the authors propose Learning Across the Noise Spectrum (LANs) and RL strategy that exposes the agent to noisy obesrvations during training to improve robustness at execution time. The authors provide experiments on modified MuJoCo environments, comparing LANs against other baselines (including a denoising autoencoder approach and a gaussian shifts approach).

**Strengths:**

The problem addressed seems relevant. The experimental protocol seems solid, with the authors performing multiple runs and reporting the standard deviation of the results. The experimental results seem reproducible. The theoretical part of the work seems relevant and interesting (even though I did not check the math carefully).

**Weaknesses:**

- Soundness: I am not fully convinced by some of the claims of the authors, particularly the one regarding the fact that "AMDPs are not a special case of POMDPs". See my questions below for more details on this matter.

- Clarity: the clarity of the paper needs to be further improved. In particular, in Sec. 2 some terms were introduced without being connected to the previous discussion. I believe the text can be polished for a smoother reading. (See my questions below for more details on this matter).

- Originality: I am not very familiar with the field of RL for asynchronous signals. Is this the first work proposing to address such a matter? For example, aren't the works [1,2,3] somehow related? I would expect the authors to provide a discussion of these related works in Sec. 4 of the document, but it appears that the authors focused their attention on other areas. Maybe these works differ from the case considered by the authors; nevertheless, if that is the case, I think it is important that this is clearly explained in the document. The discussion in Sec. 4 feels rather too general.

- Significance of the empirical results: in 3 out of the 5 tested environments the proposed method (LANs) performs similar to the other baselines. I believe this is not a reason to reject the paper alone; still, I believe the results to be a bit "mixed".


[1] - http://ieeexplore.ieee.org/document/1193736

[2] - https://arxiv.org/abs/2309.11096

[3] - https://arxiv.org/abs/2203.12759

**Questions:**

- line 113: What is the symbol after "S=". Is it a set product? Please clarify this in the text. Also, in line 122 the state space is defined using a different notation. Please unify.
- line 129: Why is the history of the k most recent actions included in the observation? I feel like this kind of information, if needed, is up to the agent to track it, no?
- line 141: "(...), we assume that the transition dynamics \tilde{P}_a" in the observation space (...)" - what do you mean by transition dynamics in the obesrvation space? Transition dynamics are usually defined with respect to states, not observations. Also, $\tilde{P}_a$ is not part of the definition of an AMDP.
- "AMDPs are not a special case of POMDPs" - I am not convinced regarding this claim. In particular, why can't we create an "extended POMDP" where we also keep track of the $\delta$'s in the state and then define the transition function to encode both the transitions between original states, as well as the $\delta$'s. The transition function would actually be similar to (2) but where we replace the observations with the new states. Then, we can come up with a stationary observation function that just presents the relevant information to the agent at each timestep depending solely on the underlying state of the POMDP. Isn't this the case?
- line 166: "well-defined rate $\lambda$" - what do you mean by a well-defined rate? While reading the article it is the first time this word comes up. How does this relate to $p(\delta)$'s? Is it that $p(\delta) = \lambda$ for every $\delta \in \mathbb{N}.$
- Eq. (7): Is $\mathbb{V}$ the variance? Please clarify it in the text.
- Sec. 5.3: Why are the results for HalfCheetah different?
- Fig. 2 Reacher: why are the rewards decreasing across time for the baselines? This is very weird, as it seems the methods are unlearning across time.

Minor comments/typos:
- line 21: "We prove that LANS acts a time-aware regularization term (...)" -> "We prove that LANS acts as a time-aware regularization term (...)"?
- line 122: "signaled space" -> "signaled state space".
- the sum range notation in summations (4) and (5) is not consistent.
- line 338: "reprsentations".
- line 356: "benchamrk".

---

### Official Review · Reviewer_hLmG · 2025-11-01

**Soundness:** 3
**Presentation:** 3
**Contribution:** 3
**Rating:** 6
**Confidence:** 3

**Summary:**

The paper models asynchronous environments as noise-parameterized POMDPs, and introduces a novel LANS approach that regularizes policies, and validates their claims through experiments. They also introduce new asynchronous MuJoCo tasks for this.

**Strengths:**

Asynchronous MDPs eliminate the assumption that agents always receive observation information without time delay, or at a defined frequency, which has significant relevance to real life problems. The paper relates these complex environments to a tractable subclass of POMDPs. The problem tackled in this paper is an important one and well motivated by the author(s), and also the introduction of novel benchmarks for evaluating these environments is significant for the community. The primary contribution of the paper is the novel LANS algorithm (which leverages these parameterized family of POMDPs), is also technically sound and experimentally validated. The related literature provided by the author is also thorough.

**Weaknesses:**

It would be nice to see the performance difference of the algorithm while varying only the average signal ratio (this is just a suggestion).
Section 5.3 (line 474) could use more explanation on why in this case the algorithm does not yield policy with lower curvature.

**Questions:**

The experiments show that LANS outperforms competing algorithms in most cases (with the exception of Hopper). Can you comment a bit more on why this is, and what this "complexity" (line 440) might imply for real life applications, and what are the limitations that could be expected?

---

### Official Review · Reviewer_LpWU · 2025-11-01

**Soundness:** 2
**Presentation:** 3
**Contribution:** 2
**Rating:** 4
**Confidence:** 3

**Summary:**

This paper introduces a parameterized family of environments called “Asynchronous MDPs”, which simulate the asynchronous arrival of sensor readings one might find in domains such as robotics. Using this formalism, the authors develop a learning algorithm called LANS that injects extra sensor delays into the learning process as a way to better handle missing sensor data. The authors prove their noise injection regularizes the model over the time dimension of the data, making it better able to estimate future value in the original noise regime. The authors plan to release set of benchmarking environments for the tasks described in this paper.

**Strengths:**

The formalism provided seems to match real-world problem settings well. Notwithstanding the inconsistencies presented below, the paper is clearly written and the formalism is well-described. The regularization theory result is a strong point of the paper, and noise regularization over time is a reasonable and intuitive approach to robustness. The connection between asynchronous sensors and noise regularization is also a helpful one.  The result in Figure 3 is a good start for understanding the regularization effect of this method.

**Weaknesses:**

This is a useful algorithmic framing and a reasonable starting algorithm, however I do think there are weaknesses in the theoretical presentation as well as empirical demonstrations.

The authors frame AMDPs as not fitting into the POMDP problem class, however this is not true with the appropriate definition of "state". A “state space” must be constructed such that observations and state transitions are both Markov when conditioned on state. Therefore, this is a POMDP, but the “state space” is “S x \delta”, and the transition includes p(\delta) as well. Given that I think this result can be framed as a POMDP with an appropriate definition of state space, section 2.2 is confusing to me as well, because they are a subset of POMDPs with the appropriate choice of state space.

A related error: In Eq 1, you write a transition function in terms of observation space but as written this is not well-defined, since it depends on the underlying non-null state.

I am unclear on one aspect of the experimental results. Are the DAE results based on the recurrent baseline, or an MLP? If the latter, this is not an appropriate baseline as the problem is partially observable.

I think the empirical results could have been more geared towards demonstrating asynchronous robustness. For example, results that swept over asynchronous-ness and show the other methods are more strongly affected. Also, generally, neither the results in figure 2 or 3 demonstrate clear superiority in the settings provided.

I am not very familiar with the literature on handling sensor delays, however I do know that it exists. This seems like a large missing area in the “related work” section. Since AMDPs are your contribution I understand there may not be a DL algorithm that exactly handles this situation, however an appropriate review of relevant techniques from asynchronous sensing would strengthen this paper.

**Questions:**

* Are the DAE results built on an MLP or an RNN?
* Can you justify/rebut my understanding of how to frame this question as a POMDP?
* Have you observed whether the learned policies are more robust to stale observations beyond the curvature measurements in Figure 3?
* Is there related work on handling asynchronous sensing that you feel would inform a reader of this paper?

---

### Meta-Review · Area_Chair_nU7E · 2025-12-30

**Summary:**

The paper tackles a relevant problem in RL, with a suitable approach. However, the reviewers raised concerns about the coverage of the related work section,  the connection with POMPDs, and, most critically, the strength of the empirical evaluation.

Given the lack of rebuttal, it is clear that the paper was not ready for publication.

**Reviewer Concerns:**

No rebuttal, therefore, the reviewers' concerns remain open.

**Reviewer Scores:**

- `LpWU`: 4 -> 4
- `hLmG`: 6 -> 6
- `CuFk`: 2 -> 2

---

### Decision · Program_Chairs · 2026-01-26

Reject